# Impact of Red Imported Fire Ant Nest-Building on Soil Properties and Bacterial Communities in Different Habitats

**DOI:** 10.3390/ani13122026

**Published:** 2023-06-18

**Authors:** Longqing Shi, Fenghao Liu, Lu Peng

**Affiliations:** 1Rice Research Institute, Fujian Academy of Agricultural Sciences, Fuzhou 350018, China; shilongqing@faas.cn; 2State Key Laboratory of Ecological Pest Control for Fujian-Taiwan Crops, Institute of Applied Ecology, Fujian Agriculture and Forestry University, Fuzhou 350002, China; l1099305602@163.com; 3Fujian Provincial Key Laboratory of Insect Ecology, Fujian Agriculture and Forestry University, Fuzhou 350002, China

**Keywords:** eusocial insect, invasive species, pH value, heavy metal, Actinobacteria

## Abstract

**Simple Summary:**

The red imported fire ant (*Solenopsis invicta* Buren) is an invasive species with a wide distribution range in China. Red imported fire ants nest in soil, have strong territorial behavior and aggression, and can cause harm to humans and livestock through their stinging and venom. This study examined the effects of red imported fire ant nests on soil properties and microbial communities in five different habitats. In each habitat, compared to the non-red imported fire ant nest control soils, pH and nitrogen content significantly increased in red imported fire ant nests, whereas the content of heavy metals, such as Cr, Pb, Cu, and Ni, significantly decreased. Although the composition, abundance, and diversity of bacterial communities of the control soils varied between habitats, bacterial community composition in red imported fire ant nest soils was similar. The abundance of Actinobacteria significantly increased in the nest soils of red imported fire ants. We believe that red imported fire ants deliberately increase the abundance of Actinobacteria to reduce the harm cause by pathogenic bacteria in the nests. This study provides reference data for the changes in soil physicochemical properties and microbial communities caused by the adaptations of red imported fire ants to different habitat soils.

**Abstract:**

The red imported fire ant (*Solenopsis invicta* Buren) is a highly adaptable invasive species that can nest and reproduce in different habitat soils. We aimed to explore the adaptability of red imported fire ants in different habitats by analyzing changes in the physicochemical properties of nest soils and bacterial communities. Five habitat types (forest, tea plantation, rice field, lawn, and brassica field) were selected. The results showed that the pH of the nest soils increased significantly in all five habitats compared to the control soils of the same habitat. A significant increase in nitrogen content was detected in the nests. The Cr, Pb, Cu, and Ni levels were significantly reduced in the soils of the five habitats, due to nesting activities. Analysis of the composition and diversity of the soil microbial community showed that, although the richness and diversity of bacteria in the nest soils of red imported fire ants in the five habitats varied, the relative abundance of Actinobacteria significantly increased and it emerged as the dominant bacterial group. These results indicate that red imported fire ants modify the physicochemical properties of nest soils and bacterial communities to create a suitable habitat for survival and reproduction.

## 1. Introduction

The red imported fire ant (*Solenopsis invicta* Buren) (RIFA) is a harmful pest, native to the Paraná River basin in South America. In the early 2000s, it invaded the southern regions of Taiwan, Guangdong, and Fujian in China and rapidly spread throughout the country [1,2,3,4]. These eusocial insects build their nests in soil, with mature nests rising up to 30–40 cm above the ground. The defensive area of RIFAs is proportional to the colony size, and adjacent populations’ size and density can influence it [5,6]. They are extremely aggressive and territorial; once a nest is threatened, large numbers of worker ants will quickly move out, stinging the intruder and releasing their venom. When humans or livestock are stung by RIFAs, they experience a piercing pain. Allergic individuals may develop symptoms such as pus-filled blisters, itching, anaphylactic shock, and even death in severe cases [7,8,9,10]. By feeding on crop seeds, fruits, young shoots, stems, and roots, RIFAs can cause a considerable reduction in crop yield, as well as in the abundance and diversity of native species (including ants) through predation and competition for resources [11,12].

As an invasive species, the RIFA demonstrates high adaptability to various habitats, as demonstrated by its nesting activities. Nest-building can affect soil properties such as physico-chemistry, microclimate, elemental ratios, and microbial communities, which can regulate material cycling and energy flow for better survival [13,14]. It has been proved that RIFA nesting and foraging activities can alter soil structure and nutrient accumulation, ultimately affecting soil physical properties and nutrient status [15]. RIFA has a diverse diet; foraging mainly occurs during the daytime, when it preys on many invertebrates (mainly insects) and feeds on crops such as vegetables, fruits, and seeds [16,17]. The consumption, digestion, excretion, and decay of residual matter can affect soil properties and microbial communities within the nest. However, the availability of food varies considerably between habitats, particularly in farmland habitats, which could lead to varying effects on soil properties in ant nests.

This study examines the changes in soil properties and microbial communities in RIFA nests in five different habitats (forest soil (FS), lawn soil (LS), tea plantation soil (TS), rice field soil (RS), and brassica field soil (BS)) under similar climatic and geographical conditions. Our study aims to provide data to better understand the mechanisms by which RIFAs adapt to different habitats.

## 2. Materials and Methods

### 2.1. Experimental Materials

To investigate the impact of RIFAs on soils with different vegetation cover, separate surveys were conducted in Lianjiang County, Fuzhou City, Fujian Province, China. The surveys were carried out in forest (26.264855° N, 119.364387° E, 68.8 m above sea level), lawn (26.268366° N, 119.366653° E, 40.8 m above sea level), tea plantation (26.252602° N, 119.404141° E, 90.2 m above sea level), rice field (ridge soil, 26.265288° N, 119.363672° E, 51.5 m above sea level) and brassica field (*Brassica* vegetables, 26.271521° N, 119.367144° E, 51.9 m above sea level). For each habitat, three large, fully formed RIFA nests (within 10 m in diameter) were located. These fifteen nests from the five habitats had a height greater than 10 cm and a diameter greater than 25 cm. Soil samples were obtained by drilling 15 cm from the top of the mound using a soil sampler. The soil samples were sieved through a 50-mesh sieve to remove individual RIFAs and other debris. The fine soil after sieving (>200 g) was collected in self-sealing bags and stored on dry ice. Three additional soil samples were taken from each RIFA nest at each habitat. Six (3 + 3) soil samples were also collected from each area at a distance of 3 m from each nest, using the same method (Figure 1).

### 2.2. Soil Physicochemical Analysis

Potential of hydrogen (pH): Ten grams of the soil sample was placed into a 50 mL beaker. Thereafter, 25 mL of water was added, stirred vigorously with a glass rod for 2 min, left for 30 min and tested with a benchtop pH meter (FF28; Mettler Toledo, Shanghai, China). Each sample was tested thrice and the average was taken.

Heavy metals: Cr, Ni, Cu, Zn, As, Cd, and Pb in soil samples were determined using aqua regia extraction and inductively coupled plasma-mass spectrometry. Thereafter, 0.1000 g of accurately weighed air-dried soil sample was placed through a 100-mesh sieve into a PTFE sealed digestion tank and wetted with five drops of ultrapure water. After 6 mL of aqua regia was added, the mixture was shaken well and allowed to stand for 10 min before the lid was secured and microwave digestion was carried out. After microwave digestion, the digestion solution was allowed to cool down before transferring to a 50 mL colorimetric tube. The digestion tank was washed three times with 1% nitric acid solution, filled up to the marked line, and mixed well. Filtration was carried out and the filtrate was analyzed for heavy metals using an inductively coupled plasma-mass spectrometer (iCAP RQ; Thermo Fisher Scientific, Waltham, MA, USA).

The Hg content was analyzed using atomic fluorescence spectrometry. First, 0.2500 g of the air-dried sample was weighed accurately through a 100 mesh sieve into a 50 mL stoppered cuvette and moistened with eight drops of ultrapure water. In a fume hood, 10 mL of aqua regia was added and the mixture was shaken well with a stopper for 2 h in a boiling water bath. Next, 10 mL of the preservation solution (0.5% potassium dichromate) was added immediately after cooling. The volume was adjusted to 50 mL with diluent (0.2% potassium dichromate), shaken well, and left to stand. The supernatant was collected and analyzed for Hg using an atomic fluorescence spectrometer (AFS-230E; Beijing Haiguang Instruments Co., Ltd., Beijing, China).

Nitrogen, phosphorus, and potassium: The total P and K contents of the soil were determined by inductively coupled plasma-mass spectrometry. First, 0.1000 g of air-dried sample was accurately weighed through a 100-mesh sieve in a PTFE crucible and 3.0 mL of nitric acid, 1.0 mL of perchloric acid, and 5.0 mL of hydrofluoric acid were added. The crucible was covered with a lid, placed on an electric hot plate and heated in a fume hood from 130 °C to 200 °C for sample digestion. The crucible was gently shaken and heated until a large amount of white smoke appeared. The heating process continued until no more white smoke was emitted, and the sample was almost dry. Depending on the residue in the crucible, 5.0 mL of hydrofluoric acid and 0.5 mL of perchloric acid may have been added and the process repeated. The crucible was cooled, 10.0 mL of 3 mol/L hydrochloric acid solution was added, and the mixture was heated until the residue was dissolved. The resulting solution was then transferred to a 50.0 mL colorimetric tube, the remaining volume was made up, the tube was shaken well, and the solution was filtered for detection.

The total nitrogen content was determined using an automated nitrogen analyzer. Approximately 1 g of the air-dried soil sample was weighed into the digestion tube, and 1 mL of potassium permanganate solution and 2 mL of sulfuric acid solution were added. The mixture was shaken well and allowed to stand for 5 min. One drop of octanol and 0.5 g of reduced iron powder were added, and the digestion tube was rotated to ensure that the iron powder reacted fully with the acid. The digestion tube was then left to stand for 45 min. Heating was stopped, and the digestion tube was allowed to cool down naturally. Then, 2 g of catalyst (potassium sulphate: copper sulfate: selenium = 100:10:1) and 5 mL of sulfuric acid were added, and the mixture was shaken well and heated at a low temperature until the reaction in the tube became moderate. The temperature was then raised to 360–380 °C. The solution and soil particles were allowed to turn gray with a slight greenish tint, after which the process was continued for 1 h. After digestion, the digestion tube was cooled before distillation. The digestion tube was then placed on an automated nitrogen analyzer (K9840; Qingdao Jingcheng Instrument Co., Ltd., Qingdao, China) for distillation and titration.

### 2.3. Extraction and Purification of Soil Microbial DNA

Total soil microbial DNA was extracted using the OMEGA Soil DNA Kit (D5625-01; OmegaBio-Tek, Norcross, GA, USA) and stored in a −20 °C freezer. The quality of the extracted DNA was determined using a NanoDrop ND-1000 nucleic acid detector (Thermo Fisher Scientific) and 0.8% agarose gel electrophoresis, whereas the extracted DNA was quantified using a UV spectrophotometer.

Primers were designed based on conserved regions in sequences using target sequences, such as microbial ribosomal, RNA that reflected the composition and diversity of the microbial communities. Diluted DNA templates were used, and specific Barcode primers, such as 338F (5′-barcode + ACTCCTACGGGAGGCAGCA-3′) and 806R (5′-GGACTACHVGGGTWTCTAAT-3′) were used for PCR amplification of the bacterial 16S V3 and V4 regions. The PCR was performed with a 25 μL reaction system (T100 gradient PCR Instrument; Bio-Rad, Hercules, CA, USA) using the following program settings: pre-denaturation at 98 °C for 2 min; then 25 cycles of denaturation at 98 °C for 15 s, annealing at 55 °C for 30 s, and extension at 72 °C for 30 s. The PCR was then extended at 72 °C for 5 min. PCR products were detected by 1.2% agarose gel electrophoresis. The PCR products were then purified and recovered using magnetic beads. The procedure was as follows: Magnetic beads (VAHTS^TM^ DNA Clean Beads; Vazyme, Jiangsu, China) at 0.8× the volume of PCR products were added to 25 μL of PCR products and shaken well to form a suspension. The mixture was adsorbed on a magnetic stand for 5 min, and the supernatant was collected. The magnetic beads were washed with 20 μL of 0.8× the volume of washing solution for the magnetic beads, and the mixture was adsorbed on a magnetic stand for 5 min. The supernatant was collected, and 200 μL of 80% ethanol was added. The PCR tube was placed in the reverse direction of the magnetic stand, and the magnetic beads were adsorbed on the other side of the PCR tube. After adsorption, the supernatant was collected. The tube was left at 26 °C for 5 s until the alcohol completely evaporated and the magnetic beads cracked. Finally, 25 μL of elution buffer was added to elute the purified PCR products, which were adsorbed on the stand for 5 min and then collected into a clean 1.5 mL centrifuge tube for storage. Quantitative fluorescence of the purified PCR products was performed using the Quant-iT PicoGreen dsDNA Assay Kit (Thermo Fisher Scientific) and a Microplate reader (BioTek FLx800; Agilent, Santa Clara, CA, USA). Based on the fluorescence quantification results, each sample was mixed with the appropriate proportion required for each sample’s sequencing volume.

### 2.4. Analysis of Soil Microbial DNA Diversity

Library preparation for sequencing was conducted using the TruSeq Nano DNA LT Library Prep Kit (Illumina, San Diego, CA, USA). The final fragments were selected and purified via 2% agarose gel electrophoresis. Prior to sequencing, quality control was conducted on the libraries with the Agilent Bioanalyzer, using the Agilent High Sensitivity DNA Kit. Qualified libraries were expected to have only one peak and no adapters. Following quality control, the libraries were quantified using the Quant-iT PicoGreen dsDNA Assay Kit (Thermo Fisher Scientific) on a Promega QuantiFluor fluorescence quantification system (Promega, Madison, WI, USA) at a concentration of 2 nmol/L or higher. The libraries were diluted in a gradient and blended in the appropriate ratio for the sequencing volume needed, then denatured to single strands with NaOH. Paired-end sequencing was performed using MiSeq sequencing. The raw sequencing data were stored in FASTQ format. Three biological replicates were designed for both the control (CK) and each treatment group.

### 2.5. Data Analysis

The data for pH, N, P, and K contents and eight heavy metals were first tested for normality, independence, and homogeneity of variance by chi-square test, followed by independent *t*-tests. The above analyses were performed by IBM SPSS Statistics software (International Business Machines Corporation, Armonk, NY, USA).

For the soil bacterial 16sDNA data, the DADA2 method [18] was applied to primer removal, quality filtering, denoising, splicing, and removal of chimeric sequences. After denoising all libraries, the amplicon sequence variants (ASVs) and ASV tables were merged, and singleton ASVs were removed. Sequence length distribution statistics were performed using an R script to analyze the length distribution of high-quality sequences across all samples. Taxonomic annotation was performed using the classify-sklearn algorithm of QIIME2 [19]. Alpha and beta diversity analysis, taxonomic composition analysis, Linear discriminant analysis, Effect Size (LEfSe) analysis, and random forest were conducted.

## 3. Results

### 3.1. Impacts of the Red Imported Fire Ant on Soil Physicochemical Properties

As shown in Figure 2, soil pH values in the five groups of RIFA nests (FS, TS, RS, LS, BS) exhibited a significant increase compared to those in the control groups. There was a highly significant increase in pH in the TS group (mean pH 6.24 in the RIFA nest compared to 4.72 in the control group (*p* < 0.001)). Soil N content in RIFA nests increased significantly in all five habitats (FS: 0.197%, *p* < 0.001; TS: 0.115%, *p* < 0.001; RS: 0.116%, *p* < 0.05; LS: 0.082%, *p* < 0.001; BS: 0.035%, *p* < 0.001). Soil P content in RIFA nests decreased significantly in all habitat groups except for the BS group (FS: 0.215 g/kg, *p* < 0.001; TS: 0.387 g/kg, *p* < 0.001; RS: 0.220 g/kg, *p* < 0.05; LS: 0.294 g/kg, *p* < 0.05). The changes in RIFA nests for soil K content were small, with no significant difference.

As shown in Figure 3, RIFA nests significantly reduced the levels of four heavy metals—Cr, Pb, Cu and Ni—in the soils of the five habitats, especially in the RS and LS habitats, with highly significant differences compared to the control (*p <* 0.001). No significant changes were observed in the levels of Cd and As in the soils affected by RIFAs. The soil Zn content was highly significantly reduced in the FS and TS habitats (*p <* 0.001), but not in the other three habitats. In the FS and RS habitats, the soil Hg content was significantly increased compared to that in the control (FS: *p* < 0.001; RS: *p <* 0.01). However, there was a significant decrease (*p <* 0.001) in the LS habitat, whereas no significant difference was observed in the other two habitats.

### 3.2. Effect of the Red Imported Fire Ant on Soil Bacterial Diversity

Through high-throughput sequencing, we obtained 1,651,413 valid sequences with a length range of 15–441 bp. Taxonomic annotation results showed that the LS–CK group had the highest number of ASVs (4220), whereas the BS–CK group had the lowest (1948) (Appendix A). Figure 4 shows the results of a comparative analysis of soil bacterial richness (Chao1 estimator) and diversity (Shannon diversity index and Simpson index) within RIFA nests in the five respective habitats. The Chao1 estimator, Shannon diversity index, and Simpson index of soil bacteria in RIFA nests in the RS and BS habitats were significantly higher (*p <* 0.05) compared to the control group. In the other three habitats, the Chao1 estimator decreased compared to the control, although there was no significant difference, whereas the Shannon diversity index and Simpson index decreased significantly (*p* < 0.05). Figure 5 displays the beta diversity of soil bacteria. There were remarkable differences in bacterial community structure among the five habitats. Furthermore, the nest-building contributed to the changes of bacterial community structure in the same habitat, especially the TS, LS and RS.

### 3.3. Effect of RIFAs on Soil Bacterial Species Composition

We compared bacterial species within RIFA nests by analyzing species relative abundance at the phylum level and mapping bacterial taxonomic trees. The dominant bacteria in soil samples from the RIFA nests and controls in all five habitats belonged mainly to Actinobacteria, Proteobacteria, Acidobacteria, and Chloroflexi (Figure 6a,c). The results showed a significant increase in the relative abundance of soil Actinobacteria in RIFA nests in all five habitats compared to the control. Among them, the relative abundance of Actinobacteria in RIFA nests was the highest in the TS, BS, and LS habitats, with a mean relative abundance of more than 44% (44.35%, 44.22%, and 45.32%). In contrast, the relative abundances of Proteobacteria and Chloroflexi in the RIFA nests in each habitat showed a decreasing trend. The relative abundance of Acidobacteria was significantly lower than that of Actinobacteria and Proteobacteria in the TS, BS, and LS habitats, and decreased further after RIFAs built their nests. However, the relative abundance of Acidobacteria was the highest of all bacterial groups in both the FS and the RS habitats (FS–CK, 27.10%; RS–CK, 28.74%). The relative abundance of Acidobacteria in RIFA nests in the FS habitat was significantly higher compared to that in the control (FS–N, 33.46%) (Figure 6a). At the genus level, the relative abundance of *Subgroup_6* in the TS, BS and LS habitats were all decreased in PIFA nests, while the relative abundance of *Nocardioides* were increased. In the control of FS and RS habitats, the genus of highest relative abundance was *Subgroup_2* (10.13% and 7.87%), which seriously increased in RIFA nests (15.97% and 8.39%) (Figure 6b).

### 3.4. Biomarker Analysis

To identify potential biomarkers of soil bacteria in the RIFA nests and controls in the five habitats, we grouped 15 RIFA nest soil samples and 15 control soil samples. We performed random forest analysis on the absolute abundance data of the genus-level classification units with 10-fold cross-validation. The top 20 ranked genera according to importance are shown in Figure 7. The top three genera were *Conexibacter* (Actinobacteria), *Smaragdicoccus* (Actinobacteria), and *Mucilaginibacter* (Bacteroidetes). All the three genera were significantly more abundant in the RIFA nest group.

## 4. Discussion

The results of this study show that RIFA nest-building activity significantly increases soil pH. These ants are known to secrete venom alkaloids on the inner surfaces of their nests, which disinfect their habitat and larval body surfaces, having antibacterial and antimicrobial effects [8,20]. The venom sac of a RIFA worker contains about 10–15 μg of venom. Up to 0.66 nL of venom is released at a time, and there can be up to 22 μg/g of venom alkaloids in the soil of a RIFA nest [9,21]. The venom alkaloids are alkaline and can affect soil pH by neutralizing acidic substances in the soil. In this study, the increase in soil pH in the RIFA nests in the TS habitat was significant (pH increased from a mean value of 4.72 to 6.24). *Camellia sinensis* L. is an acid-loving plant that grows in soils with a pH of 4.0 to 6.5, with an optimum pH of 5.5 [22]. However, in recent years, soil acidification in tea plantations has become increasingly serious. Tea trees are “acid-producing plants”, absorbing large amounts of salt-based ions and active aluminum into the soil during growth and releasing large amounts of hydrogen ions and organic acids into the soil, causing soil acidification [23,24,25]. Further research is required to investigate whether the ‘acid production’ of tea trees causes defensive behavior in RIFAs, leading to a significant increase in pH in their nests. As with most eusocial insects, dead individuals are quickly removed from the nest in order to avoid the growth and reproduction of pathogens, parasites, and other substances that harm the population and pose a direct threat to the colony [26,27]. Therefore, RIFA carcasses have little effect on the soil nutrients in the nest. In this study, soil N levels increased significantly in the nests of RIFAs in all five habitats. However, the total P and K levels showed variations, which may be related to the different types of food foraged by RIFAs. The plant species in the five habitats were different from each other, including trees (forest), shrubs (tea plantations), and grasses (rice fields and lawn). Therefore, the food types available to RIFAs in each habitat varied considerably, with varying levels of N, P, and K. The accumulation of food in nests and excretion by RIFAs caused the corresponding changes in soil N, P, and K levels [28]. In addition, the results of this study showed that the nesting activities of imported red fire ants also significantly reduced the levels of four heavy metals—Cr, Pb, Cu, and Ni—in the soil. This may be due to an increase in the pH of nest soils, which leads to a decrease in the activation of heavy metals [29]. Moreover, it has been shown that many species of ants, including imported red fire ants, can accumulate heavy metals from their surroundings [30,31,32]. We speculate that the RIFAs have accumulated heavy metals from the nest soil and are then removed from the nest when they die, thus reducing the heavy metal levels in the nest. This hypothesis needs to be confirmed by further research.

Bacteria were the most diverse and abundant microbial group in soil [33]. Soil pH is a key factor affecting the composition of bacterial communities, as most bacteria are sensitive to pH and low soil pH is not conducive to their growth, leading to a decrease in bacterial diversity and abundance [34,35,36]. In the present study, the pH values within the RIFA nests in three habitats, FS, TS, and LS, increased significantly compared to those in the CK groups (Figure 2). However, the diversity and abundance of soil bacteria showed a significant decrease (Figure 2a,b,d), which may be related to the nesting activity of RIFAs. There is evidence that venom alkaloids secreted by red fire ants have antibacterial and antimicrobial effects [37,38,39]. Since RIFAs secrete venom alkaloids on the interior surfaces of their nests to disinfect their habitat and larval surfaces [8], the amount of venom alkaloids secreted can affect the amount of bacteria in the soil to some extent. Newly mated RIFA queens can use scent to locate Actinobacteria-rich soil for nesting [40], and worker ants also prefer to expand their nests in Actinobacteria-rich soil [41]. Soil-nesting insects have a symbiotic relationship with Actinobacteria; this can produce antibiotics to protect insects from pathogenic bacteria and fungi [42,43]. Our results also show that the abundance of Actinobacteria in RIFA nests increased significantly in all five habitats and overtook Proteobacteria as the most abundant group (except in FS). Even in the FS habitat, where the abundance of Actinobacteria was low and Acidobacteria were the dominant flora (16.72% relative abundance of Actinobacteria and 27.10% relative abundance of Acidobacteria in CK), as well as the RS habitat (9.45% relative abundance of Actinobacteria and 28.74% relative abundance of Acidobacteria in CK), the relative abundance of Actinobacteria increased significantly after nesting by RIFAs (relative abundance of Actinobacteria: FS–N, 24.95%; RS–N, 26.37%). This suggests that RIFAs prefer to provide more favorable conditions for the growth of Actinobacteria when nesting.

## 5. Conclusions

In this study, we analyzed the impact of RIFA nesting on soil physicochemical properties and ecology by comparing the differences in soil pH, N, P, K and heavy metal contents and bacterial communities across five typical habitats. We proved that RIFA nesting significantly increased the soil pH, N content and abundance of Actinobacteria. The results offer new insights and data regarding how this invasive species can adapt to and thrive in different habitats.

## Figures and Tables

**Figure 1 animals-13-02026-f001:**
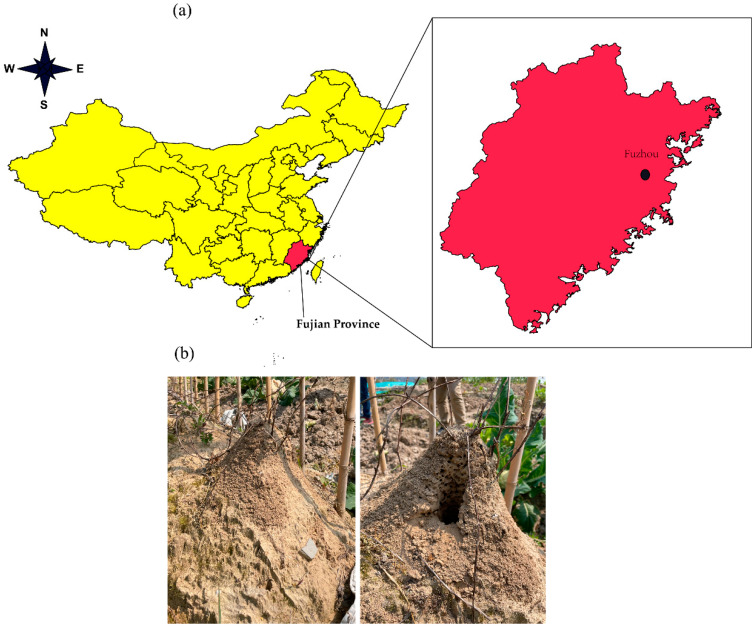
Geographical locations of the soil samples (**a**), before and after taking soil samples from the red imported fire ant (*S. invicta*) nests (**b**).

**Figure 2 animals-13-02026-f002:**
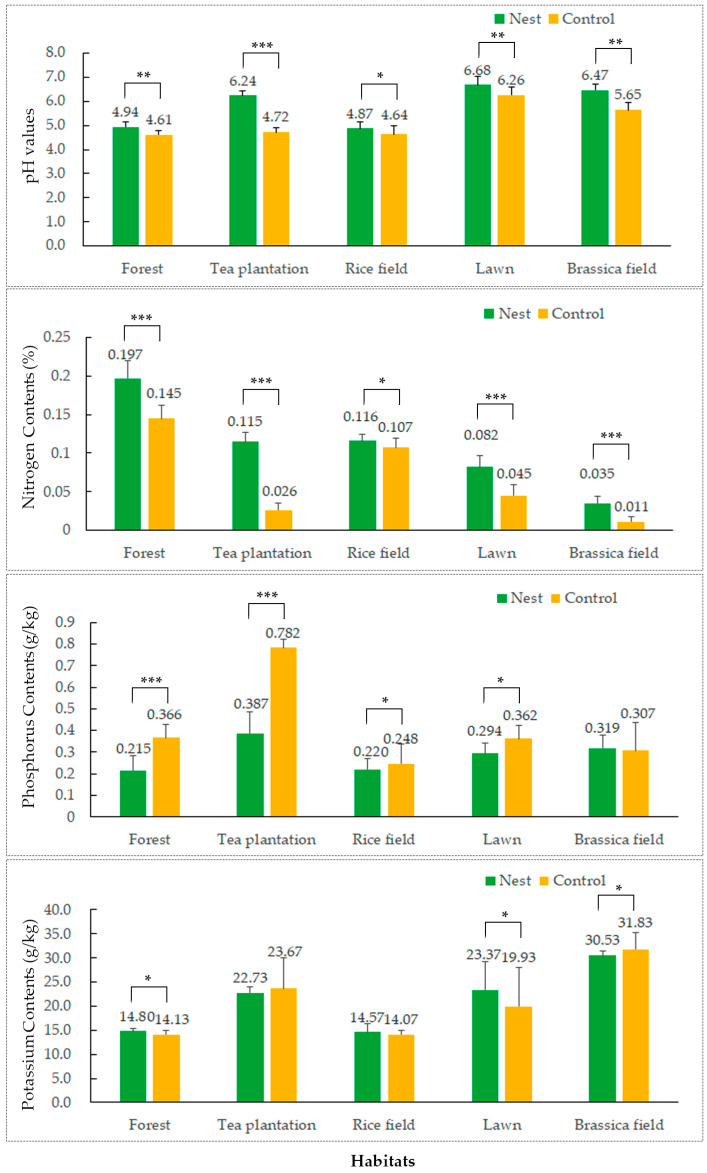
Effect of the red imported fire ant (*S. invicta*) on soil pH and N, P, and K contents. In the same habitat: independent *t*-tests, * *p* < 0.05; ** *p* < 0.01; *** *p* < 0.001.

**Figure 3 animals-13-02026-f003:**
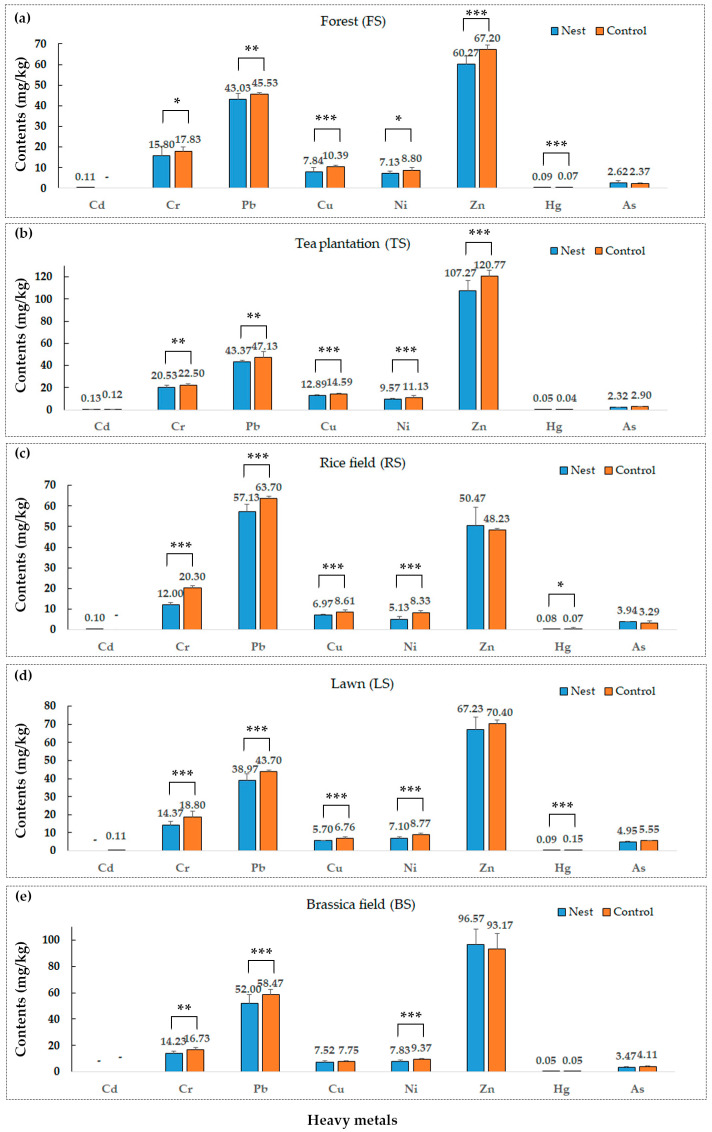
Effect of the red imported fire ant (*S. invicta*) on soil heavy metals. In the same habitat: independent *t*-tests, * *p* < 0.05; ** *p* < 0.01; *** *p* < 0.001; -: below the detection limit (<0.009 mg/kg).

**Figure 4 animals-13-02026-f004:**
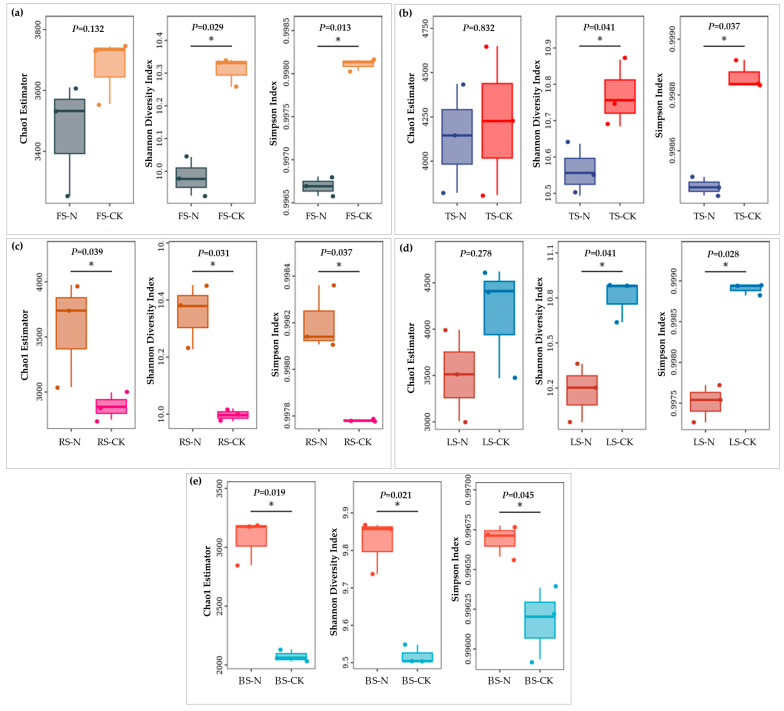
Effect of the red imported fire ant (*S. invicta*) on alpha diversity of soil bacteria. In the same box plot: * *p* < 0.05, *t*-tests. (**a**) FS, forest soil; (**b**) TS, tea plantation soil; (**c**) RS, rice field soil; (**d**) LS, Lawn soil; (**e**) BS, brassica field soil. N, collected in the nests; CK, control groups, i.e., samples collected at a distance of 3 m from the nest.

**Figure 5 animals-13-02026-f005:**
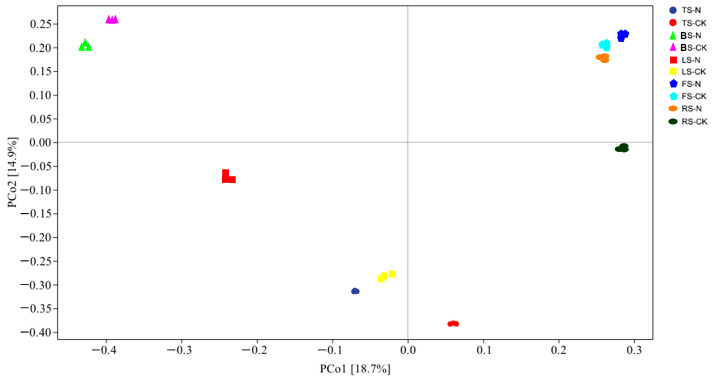
Effect of the red imported fire ant (*S. invicta*) on the beta diversity of soil bacteria. FS, forest soil; TS, tea plantation soil; RS, rice field soil; LS, Lawn soil; BS, brassica field soil; N, collected in the nests; CK, control groups, i.e., samples collected at a distance of 3 m from the nest.

**Figure 6 animals-13-02026-f006:**
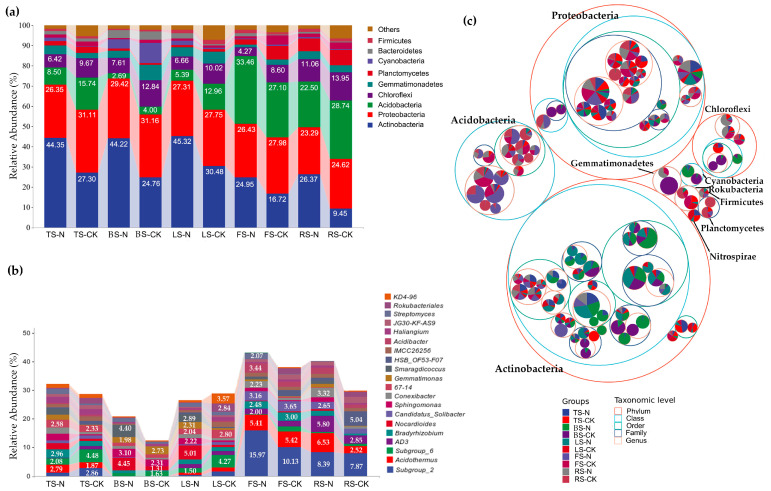
Impact of the red imported fire ant (*S. invicta*) on soil bacterial diversity and composition. (**a**) Comparison of relative abundance of bacterial phyla in the soil. (**b**) Comparison of relative abundance of bacterial genus in the soil. (**c**) Bacterial taxonomic tree. The largest circle represents the phylum, with progressively smaller circles representing the class, order, family, and genus. The innermost dots represent the top 100 amplicon sequence variants (ASVs) in abundance and their size is proportional to the abundance of the respective ASVs. Each ASV is depicted as a pie chart, showing its composition in each group. Segment size corresponds to the relative abundance of that taxonomic unit in the respective group. FS, forest soil; TS, tea plantation soil; RS, rice field soil; LS, Lawn soil; BS, brassica field soil; N, collected in the nests; CK, control groups, i.e., samples collected at a distance of 3 m from the nest.

**Figure 7 animals-13-02026-f007:**
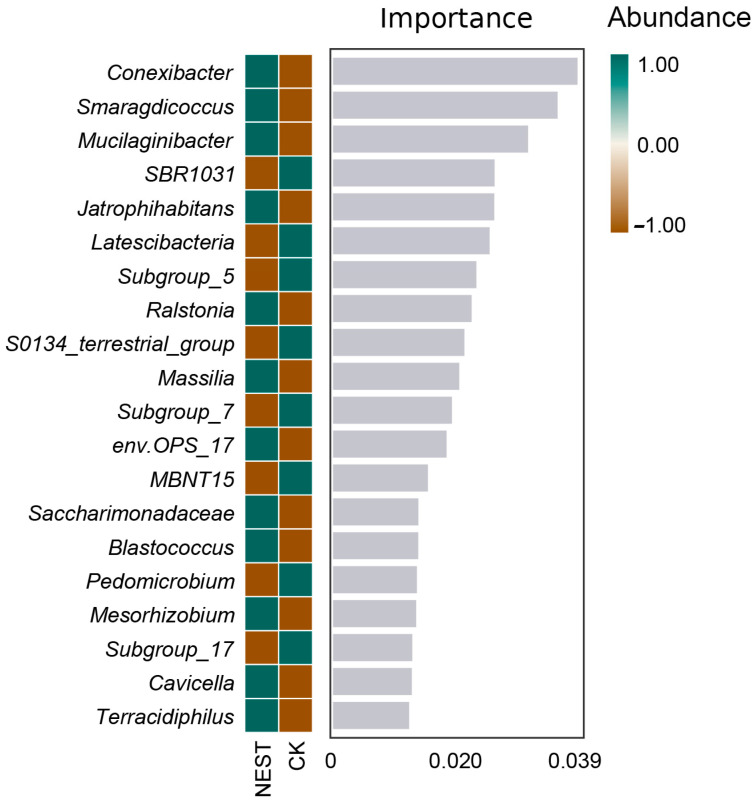
Random forest analysis of soil bacterial markers at the species level. The absolute abundance data of taxonomic units at the genus level underwent 10-fold cross-validation, and the top 20 most important genera are displayed. The bar chart presents the importance score of each genus for the classifier model on the x-axis and the genus name on the y-axis. The heatmap shows the distribution of the abundance of these genera in each group. NEST represents the mean values of all 15 samples from the red fire ant nests across the five habitats, whereas CK represents the mean values of all 15 control samples from the same five habitats.

## Data Availability

The data that support the findings of this study are available from the authors upon reasonable request.

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
