# Peer review of "Impact of Red Imported Fire Ant Nest-Building on Soil Properties and Bacterial Communities in Different Habitats"

_animals, 2023, doi:10.3390/ani13122026_

Round 1

Reviewer 1 Report

In this work, the authors compare the differences in the physical and chemical properties of the soil inside and outside the nests of RIFAs in different habitats and found that in all habitats, fire ant activities increase pH and nitrogen content, and decrease the content of heavy metals. The changes of P and K are affected by habitat type. The authors also analyzed soil microbes from different habitats and identified potential biomarkers of soil bacteria.

The data presented here are very interesting. Conclusions are helpful to understand the ecological adaptation of RIFA to the invaded areas. However, there are significant flaws in this work. The number of replicates for the microbiota experiment is not as many as regular studies. Data analysis and presentation need to be significantly improved. Moreover, I don’t see any connection between the two parts of the study. This study can be split into two papers. And additional work is needed to tell full stories. Rewriting of the introduction and discussion sections is needed to state importance of this work and contribution of data to science.

Some concerns are listed below:

1. There were only three biological replicates per habitat in the analysis of soil bacterial diversity. There should be 5-10 replicates for soil bacterial diversity analysis.

2. Is it appropriate to use bacteria as biomarkers? Are there any related reports in literature? The significance of relevant content of biomarkers should be discussed.

3. Are the differences in bacterial communities inside and outside the nest and between different habitats related to changes in physicochemical properties?

4. Table 1. The number of decimal places is uneven. the standard deviation or standard error is not indicated. The aesthetics need to be strengthened.

5. P-values ​​in statistical analyzes should be “P < 0.05”.

6. A figure is preferred to present data in Table 2.

7. Be consistent with same color for same treatment across Fig. 2, Fig. 3 and Fig. 4b.

8. It is very hard to distinguish different treatments in Figure 3. Use same color for different treatment from the same habitat, and differenttypes of marker to distinguish different treatment.

9. Supplementary Table 1 is missing.

10. Fig 4b. Do the Latin names in the figure indicate the largest phyla? What does the size of the pie chart mean? Proportion? Also, check if Latin names require italics.

11. Fig 4 shows the variation of soil microorganisms at the phylum level in different habitats, and the biomarker analysis shows the overall dominant genera at the genus level. However, the differences at the genus level across different habitats are lacking.

moderate. significant improvement is needed.

Reviewer 2 Report

The topic covered in the article is interesting and falls within the scope of the Journal. The manuscript is well-written and organized. As far as I could evaluate the data generation, their analysis is straightforward and appropriate and supports the authors' conclusions. The text is clearly written and straight to the point. Therefore I recommend this article for publication.

Reviewer 3 Report

I have read the manuscript entitled “Impact of Red Imported Fire Ant Nest-Building on Soil Properties and Bacterial Communities in Different Habitats” by Longqing Shi and colleagues, submitted to the Animals magazine. The Authors studied effects of the red imported fire ants (Solenopsis invicta Buren) on soil properties and microbial communities in China. They found that the ant nest significantly increased pH and nitrogen content, but the content of heavy metals decreased. Generally, the results are interesting. As the ant species is invasive, such data could be interesting for the wired audience. However, I think the version of the manuscript could be improved. I hope that the following comments can be used to improve it.

I believe, that information about the research design and statistical analyses, as well as the result presentation, could be improved.
Authors collected data on the area (line 82) “where large, fully formed RIFA nests were located.”, and (line 87-89) “Three additional soil samples were taken from each RIFA nest at each location. Three soil samples were also collected from each area at a distance of 3 m from the nest, using the same method (Figure 1).” However, I found no information how many ant nests in each of the five areas were used in the study. Additionally, in the results section, for the results of statistical analyses, there are no information on the sample sizes. Typically, only the P value is presented, without information on degree of freedom (df), neither the value of used statistical test (t value, for t Student test). The information are crucial for the results interpretation.

lines 240-243: “To compare the differences in microbial diversity between habitats, we calculated the Bray–Curtis distance matrix using the ASV abundance data of the RIFA nests and the controls in the five habitats and performed principal coordinate analysis (PCoA).” – it is rather part of Materials and method section, not Result section, I think. 

lines 193-94: “The data of pH, N, P, and K contents and eight heavy metals was firstly tested for normality, independence, and homogeneity of variance, followed by independent t-tests.” More precise information is recommended, i.e. which test was used to check the assumption on normality and homogeneity of variance. However, for me, the most important is how “independence” was tested.

lines 213-214: “The effect of RIFA nests on the soil K content was small, with no significant increasing or decreasing trends.” I am not sure if it is good to write about “increasing or decreasing trends”, when the differences (nest vs. control areas) were tested, and the differences were not significant.

Table 1 and Table 2. “In the same habitat: * P ≤ 0.05; ** P ≤ 0.01; *** P ≤ 0.001”. If the results of numerous t Student test are presented here? If yes, it should be stated. Additionally, is such situation, the Bonferroni correction for the multiple comparisons should be used, I believe.

Table 2. Why the value of Hg for nests located at Rice field is presented with another precision? (0.0803 vs. 0.072 for control).

Figure 2. Show the precise P values, please – the information “* ≤ P . 0.05” is not enough.
Please forgive me for my possible misunderstanding, but I think that the used test (i.e. the Dunn’s test) is a’posteriori test – once initial ANOVA has found a significant difference in three or more groups, Dunn’s test can be used to find which specific means are significant from the others. However, I found no information on using ANOVA in your study.
Additionally, what is presented on the graphs? Is it average value, standard error and standard deviation (like typically for data with the normal distribution), or median and quartiles?

Thus, again: I believe, that information about the research design and statistical analyses, as well as the result presentation, could be improved.

Figure 3. should be improved. In the present form is difficult to read and to understand. The information in the text (lines 243-245) “Figure 3 shows that the differences in bacterial community structure were significant not only between habitats, but also between RIFA nests and controls within the same habitat.” is not enough to understand the figure.

Figure 4a. The legend should be arranged in reverse order, as on the Figure Actinobacteria is at the bottom, and „Others” on the top.

Figure 6. I do not understand the legend, i.e. ‘Abundance’, as each of the genera is marked “blue” or “red” colour. If the figure (i.e. colures) is correct?

Generally, I think that legends of figures and tables could be prepared in a better way. In scientific papers, figures and tables should be ‘self-explanatory’, i.e. it should be possible to understand a figure /a table, having the figure/the table and the legend only (but without looking for more information in the text of the paper). In the present form, there are a lot of abbreviations, and it could be confusing for readers. Even short information that – for example Figure 2 – “N” = samples collected in the nests, “CK” = control groups, i.e. samples collected at a distance of 3 m from the nest, and “FS, TS, RS, LS, VS” = different habitats, would be really useful for readers.
I believe, that in the legends, using name of the ant [‘the red imported fire ant (Solenopsis invicta)’] instead of the abbreviation “RIFA” would be better.

Reviewer 4 Report

The manuscript, "Impact of Red Imported Fire Ant Nest-Building on Soil Proper-2 ties and Bacterial Communities in Different Habitats" provides new information regarding the impact of fire ant colonies on soils in different types of agricultural and non-agricultural habitats. 

Overall, the manuscript is well written. Edits to improve style and clarity are made in the attached pdf. There are a few issues that must be addressed before the manuscript is ready for publication. 

1. There are some additional details needed in the methods section. Specific comments regarding details can be found in the pdf. The most important detail needing clarification is the number of colonies that were sampled. The way the section is currently worded, it sounds like 4 sub-samples were collected from each mound in each soil type, and then 3 additional Check samples were taken. Is this meant to say that 3 different Nests were sampled at each soil type site? If only 1 Nest per site was analyzed, that limits the interpretation of the results, since the soil-samples are just sub-samples of the same Nest. If they were separate Nests, please provide information on the average distance between sampled Nests. 

2. 'vegetable field' is used as a category of soil-type. I recommend being more specific here and calling this Brassica field. Since plants in this group are regularly used for biofumigation of soils to alter microbial communities, it is important to discuss this issue specifically, as it relates to how soils growing these plants may differ (or not differ) from other soil types. 

Figure 3. The small size of the dots and many colors make the image difficult to interpret. If possible, change the shape of the dots for Nest vs. Check (maybe circle vs. triangle) and then have each color be a different soil type. Also, enlarge dots if possible. 

Figure 4. Can this figure take up a whole page? It is difficult to see the small circles. Also, maybe keep each soil type the same color but have Next be solid and the Check be striped or patterned differently.  

Round 2

Reviewer 1 Report

There are already several studies on bacterial communities of the fire ant mound soil. I just noticed a new publication on the same topic. Bamisile, B.S.; Nie, L.; Siddiqui, J.A.; Ramos Aguila, L.C.; Akutse, K.S.; Jia, C.; Xu, Y. Assessment of Mound Soils Bacterial Community of the Red Imported Fire Ant, Solenopsis invicta across Guangdong Province of China. Sustainability 2023, 15, 1350. https://doi.org/10.3390/su15021350. Although the manuscript has some publishable data, the merits are low to science. The authors did not respond to the comment that there seems no connection between the two parts.

moderate.

Author Response

Piont 1: There are already several studies on bacterial communities of the fire ant mound soil. I just noticed a new publication on the same topic. Bamisile, B.S.; Nie, L.; Siddiqui, J.A.; Ramos Aguila, L.C.; Akutse, K.S.; Jia, C.; Xu, Y. Assessment of Mound Soils Bacterial Community of the Red Imported Fire Ant, Solenopsis invicta across Guangdong Province of China. Sustainability 2023, 15, 1350. https://doi.org/10.3390/su15021350. Although the manuscript has some publishable data, the merits are low to science. The authors did not respond to the comment that there seems no connection between the two parts.

Response 1: Thanks so much for the reminding. In this manuscript, we focused the influence red imported fire ant nest-building soil properties and bacterial communities. Therefore the correlation analysis between bacterial communities and physicochemical properties was excluded. However, we will try to analyse the correlation in further study, thanks again for your kind reminding.

Reviewer 4 Report

The revision of the manuscript has been approved for clarity. My only additonnal suggestions is that the authors change Figure 2 y-axis from Kalium to Potassium, since they made that change in the text. After that change, the manuscript should be ready for publication.

Author Response

Point 1: The revision of the manuscript has been approved for clarity. My only additonnal suggestions is that the authors change Figure 2 y-axis from Kalium to Potassium, since they made that change in the text. After that change, the manuscript should be ready for publication.

Response 1: Thanks so much for the reminding. It has been corrected.